# Neuromorphic-Based Neuroprostheses for Brain Rewiring: State-of-the-Art and Perspectives in Neuroengineering

**DOI:** 10.3390/brainsci12111578

**Published:** 2022-11-19

**Authors:** Michela Chiappalone, Vinicius R. Cota, Marta Carè, Mattia Di Florio, Romain Beaubois, Stefano Buccelli, Federico Barban, Martina Brofiga, Alberto Averna, Francesco Bonacini, David J. Guggenmos, Yannick Bornat, Paolo Massobrio, Paolo Bonifazi, Timothée Levi

**Affiliations:** 1Department of Informatics, Bioengineering, Robotics System Engineering (DIBRIS), University of Genova, 16145 Genova, Italy; 2Rehab Technologies, Istituto Italiano di Tecnologia, 16163 Genova, Italy; 3IMS Laboratory, CNRS UMR 5218, University of Bordeaux, 33405 Talence, France; 4Department of Neurology, Bern University Hospital, University of Bern, 3012 Bern, Switzerland; 5Department of Rehabilitation Medicine, University of Kansas Medical Center, Kansas City, KS 66103, USA; 6Landon Center on Aging, University of Kansas Medical Center, Kansas City, KS 66103, USA; 7National Institute for Nuclear Physics (INFN), 16146 Genova, Italy; 8IKERBASQUE, The Basque Fundation, 48009 Bilbao, Spain; 9Biocruces Health Research Institute, 48903 Barakaldo, Spain

**Keywords:** closed-loop, electroceuticals, in vitro, in vivo, neuromorphic, real-time, hybrid neurotechnologies, brain rewiring

## Abstract

Neuroprostheses are neuroengineering devices that have an interface with the nervous system and supplement or substitute functionality in people with disabilities. In the collective imagination, neuroprostheses are mostly used to restore sensory or motor capabilities, but in recent years, new devices directly acting at the brain level have been proposed. In order to design the next-generation of neuroprosthetic devices for brain repair, we foresee the increasing exploitation of closed-loop systems enabled with neuromorphic elements due to their intrinsic energy efficiency, their capability to perform real-time data processing, and of mimicking neurobiological computation for an improved synergy between the technological and biological counterparts. In this manuscript, after providing definitions of key concepts, we reviewed the first exploitation of a real-time hardware neuromorphic prosthesis to restore the bidirectional communication between two neuronal populations in vitro. Starting from that ‘case-study’, we provide perspectives on the technological improvements for real-time interfacing and processing of neural signals and their potential usage for novel in vitro and in vivo experimental designs. The development of innovative neuroprosthetics for translational purposes is also presented and discussed. In our understanding, the pursuit of neuromorphic-based closed-loop neuroprostheses may spur the development of novel powerful technologies, such as ‘brain-prostheses’, capable of rewiring and/or substituting the injured nervous system.

## 1. Introduction

Diseases and injuries of the nervous systems constitute about 6.3% of the Global Burden of Disease [1], affecting more than one billion people worldwide with a tendency to worsen in the next years due to the ageing of the population. This, of course, has and will have a massive impact on society and the economy. Despite significant scientific and technological development in recent years, first choice treatments still fall short of fully controlling symptoms and promoting enduring recovery [2,3].

In particular, stroke is one of the leading causes of death and disability [4]. In its acute phase, blood—and thus oxygen—supply is interrupted, leading to a cascade of neurochemical processes, including glutamatergic excitotoxicity and oxidative damage, which cause cell death and loss of connectivity [5]. As a consequence, local information processing capabilities, and information exchange between distant circuitry are impaired, thus causing disruption of the process of segregation and integration of information in the brain [6]. The final result of this disruption of structural and functional integrity of the brain, also called disconnection, is expressed as deficits of different modalities, including motor, sensory, and cognitive. Traumatic brain injury (TBI), involving not only mechanical lesions but also a variety of ensuing neurochemical processes, is another important cause leading to connection dysfunction and, thus, multimodal deficits [7].

In both cases, an aspect of major importance driving scientific research and therapeutic intervention is that many of the same neurochemical processes that lead to cell death and disconnection are also related to increased susceptibility to plastic changes [8]. Thus, the standard-of-care for post-stroke and TBI rehabilitation aims to take advantage of this plasticity window following stroke or trauma to reconstruct brain connectivity. This is usually carried out by physical therapy, which can be augmented by pharmacotherapy and/or robotic aid [9,10]. However, pharmacological treatment is still limited and very few truly innovative molecules have been discovered in recent years [11]. In fact, mainstream vascular growth strategies face high hemorrhagic risks and, at the same time, the relationship between novel antibody-based drugs and transport across the (damaged) blood–brain barrier is still obscure [8,11]. By its turn, robotic aid relies on complex and expensive equipment with limited availability, imposing the need for transportation of mobility-impaired patients to the medical facility, which can be a major obstacle for full treatment adherence. In any case, unfortunately, only a minority of stroke survivors with hemiparesis are able to achieve complete functional independence in simple activities of daily living with standard treatment. For this reason, favoring the recovery of cognitive and motor functions of patients with disabilities is a global priority in healthcare and research [12].

To this end, innovative biological and engineering approaches aimed at brain rewiring are being explored to take better advantage of improved plasticity. The biological approach of neural transplants relies on the intrinsic and increased plasticity of neuronal cells to promote structural and functional reconstruction. However, the interaction of the graft with the host nervous tissue is often poorly predictable and can yield maladaptive processes [13]. In contrast, major progress has been made on the engineering front, in the fields of bioelectronics and neural engineering, which allowed the development of electroceutical-based devices [14,15], such as neural interfaces and neuroprostheses, capable of facilitating or recovering functionality in the dysfunctional/disconnected neural tissue [1,12].

The direct interfacing of artificial circuits with large neuronal networks also spurred the development of novel ‘neurobiohybrid’ (or simply ‘biohybrid’ or ‘hybrid’) systems [16,17], which can become interesting clinical solutions for treating brain lesions [18]. Of particular interest here, neuromorphic computing systems, which are envisaged as energy-efficient and real-time capable devices for health care, are at the core of next-generation hybrid neurotechnology for brain repair in which a real-time interface between a computational device and a biological system is mandatory [19,20]. A roadmap on neuromorphic computing demonstrating that current and future neuromorphic systems will be able to process and dynamically learn signals at a low power consumption has been recently described [21]. It naturally follows that complex processing of the biological signals could then be embedded in new types of neuroprostheses.

In this paper, after providing supporting definitions of neuroprosthetics and (bio/neuro) hybrid systems (Section 2), we review the state-of-the-art neuromorphic-based neuroprostheses for brain repair (Section 3). We start by in-depth revisiting the pioneering work of the authors in the development of an innovative bi-directional neuromorphic prosthesis tested on in vitro assays (Section 3.1) and then proceed with describing other scientific and technological progress grouped by the different elements that characterize the complete system, namely the neural interface (Section 3.2), the neural processing (Section 3.3), and the biological component (Section 3.4), this last one in a translational perspective. For each of these system components, we also highlight the main advancements and challenges. The focus will be on closed-loop approaches that allow better synergy between technological and biological counterparts, enhancing their function as a true brain prosthesis. Emphasis will also be on stroke and connectivity dysfunctions, with neuromorphic technology applied to induction of plasticity, and thus brain rewiring, given the accumulated experience of the authors and more widespread literature. However, such topics are not comprehensive of the field, which can be extended to other brain injury scenarios. We will put forward such perspectives whenever they are due.

## 2. When Neuroscience Meets Engineering

### 2.1. Neuroprosthetics: A Definition

Uncovering the underlying mechanisms of the brain in order to promote functional rewiring is a truly challenging goal, given that the interactions between its elements occur not only at various complexity levels (from molecular to system level) but also at distinct temporal (from sub-millisecond to days and years) and topological scales (from single node to the network as a whole) [22]. These challenges are being met by the development of the ‘neural engineering’ field (or ‘neuroengineering’), in which the nervous system is interfaced with electro-electronic equipment (neural interfaces) for bi-directional transfer of information in the pursuit of scientific investigation, novel treatments, or even for human enhancement [23,24].

The dual interaction between neuroscience and neurotechnologies, which can be depicted as a ‘tale of two loops’ [25] (Figure 1a), resulted in the development of peculiar types of neural interfaces named ‘neuroprostheses’. As reported in [26], a neuroprosthesis can be defined as ‘a device or a system that has an interface with the nervous system and supplements or substitutes functionality in the patient’s body’. Typically, neuroprostheses are mostly intended to restore sensory capabilities [27,28] or motor functions [29]. In a few cases, researchers tried to restore cognitive/motor capabilities by means of a direct ‘bypass’ of the brain damage [30,31,32]. In different studies, ranging from rodents [33], nonhuman primates [34], and humans [35], memory encoding was induced via neuroprostheses.

### 2.2. Communication Modalities

Neural activity (or neural code) can be read at different levels of invasiveness, risk, and spatial and temporal resolution [36]. In the same vein, it can also be modulated (writing of the neural code) by different modalities. The easier way to deliver a stimulation is via an ‘open-loop’ approach (Figure 1b, top), in which a pattern of stimuli is freely delivered to the neuronal network, i.e., stimulus parameters are not linked to any dynamical feature of the stimulated neural element or its afferences/efferences. Although methodologically straightforward, this unconstrained modality usually incurs both low efficiency (excessive delivery of stimulation during moments in which it is not needed) and low efficacy (stimulation is not adjusted to ongoing neural dynamics) [36,37]. Closed-loop configurations can overcome both these issues. By means of locking stimulation delivery and its parameters to meaningful features of recorded brain activity, the intervention was adjusted for optimal performance, to the extent of knowledge on the disorder neurobiology. This allows for a much-desired network-dependent and subject-tailored stimulation. Closed-loop stimulation has proven to be highly effective in restoring brain functionalities, not only by artificially coupling two disconnected brain areas but also by promoting neuronal plasticity, exploiting the causal relationship between presynaptic and postsynaptic activity [38,39,40].

Inherently, such an approach implies an online processing of the neuronal signal to extract meaningful electrographic features, including triggering events (Figure 1b, bottom). Different strategies are suitable for recording (brain reading) in the closed-loop architecture. One alternative is based on the recording of multi-unit or single-unit activity in the form of extracellular spikes, which occupy the high spectral range of the electrophysiological signal. In this case, stimulation can then be triggered based on the occurrence of a single spike, a combination of multiple spikes falling in a pre-defined time window, highly packed spikes from a single neuron (burst), or a network burst detected on multiple recording channels (Figure 1c bottom right). Additionally, local field potentials (LFPs), on the low-frequency range of the spectrum, carrying information from a larger volume and thus more informative of network activity, are also used to trigger stimulation. In this case, several electrographic features, including the power or the instantaneous frequency, phase, measurements of synchronization across channels, detection of meaningful electrographic signatures (e.g., epileptiform spikes, thalamocortical spindles), etc., may serve as an input control signal for stimuli (Figure 1c bottom left).

### 2.3. Biohybrid Systems

A major challenge in designing novel bi-directional neuroprostheses is the complexity of the natural interactions among different brain areas in vivo and how this translates into the development of accurate stimulation protocols, as well as into forms of testing those devices [41]. This is the scenario that motivated, more than 20 years ago, researchers to start exploring the possibility to create ‘biohybrid’ systems based on the functional interaction between a biological (e.g., neural) system and an artificial device [16]. Pioneering developments have been carried out with simple biological neural networks, such as the pyloric circuit in the crustacean stomatogastric ganglion for a proof of concept [42], and later by our group using the isolated severed spinal cord of rats in order to evoke locomotor-like activity [43]. These biohybrid systems can also be characterized by in vitro cultures of neurons typically coupled to micro-electrode arrays. Such a methodological approach constitutes a simple but powerful experimental model which can be easily controlled, manipulated, and monitored [44]. Moreover, thanks to their simplicity, in vitro neuronal cultures have also been recognized as a successful model system of neuronal activity [45]. As underlined by S. Potter in [46] ‘*Thanks to their controllability and relative simplicity, artificially embodied in vitro networks provide excellent test beds for studying plasticity mechanisms. It is not hard to imagine that this electrical training and modulation of cortical tissue could form the basis of future adaptive, closed-loop Brain Machine Interfaces*’.

The route indicated by Potter, one of the most prominent pioneers in the use of in vitro cultures for biohybrid experiments, can be named ‘translational methodology’. This methodological approach allows respones to basic scientific questions and to test new technological solutions, as a first step, in simple, more accessible experimental models, thus reducing the experimental variability and facilitating the interpretation of the results. As a second step, by capitalizing on the results on the first one, the complexity of the system can increase, paralleled by an upgraded version of the previously developed technology. The previously described in vitro experimental approach is the one adopted for developing a novel concept of closed-loop hybrid neuromorphic neuroprosthesis for restoring neuronal communication. This has been carried out in the scope of the European-funded BrainBow project, the results of which have been recently published [47], which we briefly revisit in Section 3.1 as a ‘case study’.

## 3. Neuromorphic Neuroprostheses: State-of-the-Art and Perspectives

### 3.1. BrainBow, a Novel Bi-Directional Neuromorphic In Vitro Sytem for Brain Rewiring

An example of innovative neuroprosthesis has been designed in the framework of the European Project ‘BrainBow’ [48], running from 2012 to 2015, and whose results have been reported in [47]. The main outcome of the project has been the realization of a novel real-time neuromorphic system (i.e., a ‘neuromorphic prosthesis’) able to artificially couple two disconnected neuronal populations in vitro (Figure 2(a1)) in a fully bi-directional way. We designed and adopted two ‘replacing’ neuroprosthetic strategies (Figure 2(a2)): *Bidirectional Bridging* (BB), to artificially replace the missing connections between two disconnected populations, and *Hybrid Bidirectional Bridging* (HBB), to replace the activity of one entire population by means of a real-time neuromorphic spiking neural network—SNN. Hereafter, we briefly report the main outcomes of the project (for a thorough description, please refer to [47]).

As anticipated, the core of the BrainBow neuroprosthesis is the design of a bidirectional hardware neuromorphic device, designed for real-time recording, processing, and stimulation of biological neurons. The whole system is composed of (i) a biological neural network, constituted by a culture of dissociated neocortical neurons plated over a micro-electrode array—MEA; (ii) an all-hardware board hosting an FPGA (Field Programmable Gate Array) to realize an energy-efficient and real-time embedded processing of biological signals and implementing a closed-loop transfer of information with the biological network. The solution to dynamically couple biological and artificial elements was inspired by the pioneering works of dynamic clamp [49,50], whose working principle is based on the alteration of the conductance of a neuron. Practically, the dynamic clamp evaluates the difference between the membrane and the reversal potentials for a well-defined conductance, multiplies this “driving force” by the desired amount of conductance, and delivers the resulting current into the neuron. Some years later the, works of Marder’s group, Le Masson and co-workers extended such an approach at the network level [42], paving the way to the possibility to put into communications artificial and biological neuronal ensembles. BrainBow worked in this direction by realizing hybrid neuronal netwoks, but without working on the conductances since it made use of abstracted neuronal models (cf. Section 3.3.2).

The experimental set-up is composed of bi-modular neocortical cultures from rat dissociated neurons (E18), constituted by two interacting populations of neurons; the physical separation between the two assemblies was achieved by means of poly-dimethyl-siloxane (PDMS)-based masks [51] which kept separate the cell bodies allowing the communication by means of bundles of neurites running across thin microchannels. Thanks to laser ablation [52], a focal lesion has been performed to physically cut the connections between the two neuronal modules. The pattern typically exhibited by in vitro cultures named ‘network burst’ [53,54] was clearly disrupted by the lesion, as the two modules started generating activity in a completely desynchronized way [47]. On the processing side (Figure 2(a1)), the system included an adaptive threshold spike detection and an extraction of network-wide events (i.e., network bursts) to (i) either send a stimulus trigger back to the culture (to implement the BB strategy) or (ii) trigger a stimulation for the SNN configured on the FPGA, which was, in turn, able to generate network-burst event able to produce triggers to stimulate back the biological culture. The neuromorphic device achieved constant sub-millisecond processing on all the recording (and stimulation) channels.

The first reconnection strategy implemented in our neuromorphic prosthesis was the Bidirectional Bridging—BB (Figure 2, panels b). The experimental design was the following (Figure 2(b1)): after the first basal phase (PreL), the lesion was executed and the spontaneous activity post-lesion recorded (PoL1), followed by BB and the PoL2 final phase. During BB, network bursts were again characterized by the involvement of the two modules (Figure 2(b2)). The cross-correlation (CC) was high during PreL, dropped to zero during PoL1, was partially restored during BB, and went back to zero during the last phase (Figure 2(b3)).

The second reconnection strategy implemented in the developed neuromorphic prosthesis (i.e., HBB) was performed in an additional set of experiments (Figure 2, panels c). The experimental design was the same adopted for the BB experiments but with the HBB phase instead of the BB one (Figure 2(c1)). During the HBB phase, the hardware-based SNN implemented in the neuromorphic prosthesis was employed to perform a hybrid bi-directional interaction. The raster plots in Figure 2(c2,c3) report what happened during a representative hybrid interaction. The biological neural network (BNN, i.e., the bi-modular network) spontaneously generated a network burst, which was detected and, as a consequence, stimulation was delivered to 10 excitatory neurons of the SNN (Figure 2(c2)). This stimulus could elicit a network burst on the SNN, thus causing a response in the form of an electrical pulse delivered back to the BNN. The opposite case (i.e., a network burst initiated in SNN) is reported in Figure 2(c3). Interestingly, the responding stimulus to the SNN was not able to elicit a good response because of the short time passed from the last network burst of the SNN: this prevents the delivery of too many stimuli in a short period of time. In terms of cross-correlation, again the results are similar to those obtained for the BB experiments: it was high during preL and was partially restored during HBB (Figure 2(c4), PoL1 and PoL2 not reported here).

The results of this original work provided clear evidence for the feasibility of a real-time (i.e., closed-loop stimulation in less than 1ms) neuromorphic prosthesis to restore neuronal communication, thus, helping to pave the way to electroceuticals, in particular in the framework of microstimulation as a therapeutic strategy for neurodegenerative diseases [55]. Several other works stemmed from this original study [21] to explore some fundamental (and critical) steps for the development of neuromorphic neuroprostheses. The main elements of innovation are related to different types of communication modalities [56] by taking advantage of the latest technological developments [57,58,59] or increasing the complexity of the experimental model [60]: all the above comes with additional opportunities and challenges, as presented in the following sections.

### 3.2. Neuromorphic Neural Interfaces

The interface between the biological and the electronic component is the key element to achieve bi-directional communication in a biohybrid system. The interface includes both the recording of neural activity and the stimulation blocks. To improve this communication, the optimal system should be precise at the single cell level but be also able to treat neural signals at a larger scale, such as the one of neural networks. In this Section, we focus on basically two types of recording and stimulation modalities, the first based on electrophysiological data and the second based on optical data, highlighting the recent developments in both fields and the related possibilities offered by neuromorphic engineering.

#### 3.2.1. High-Density MEA-Based Recording and Stimulation

When developing new generations of neuroprosthetic systems by means of in vitro preparations, a fundamental role is played by the technological substrate where neurons grow and establish complex networks. It is evident that collecting more local information by increasing the spatial sampling raises new possibilities in mapping the propagation of the electrophysiological activity, thus improving the performance of the neuroprosthesis. It was demonstrated that functional topological features of neuronal networks are strongly affected by the number of recording sites used to infer their functional connectivity [61]. In the example described in Section 3.1, each of the two modules contains about 1000 neurons/mm^2^, whose emerging electrophysiological activity is recorded by no more than 30 micro-electrodes. This means that the bi-directional interface depicted in Figure 2 is driven by a few electrodes which deeply undersample the patterns of spontaneous activity. More modern in vitro neuronal network models are designed to present three-dimensional (3D) features and more than two interconnected sub-populations (cf., Section 3.4.1), paving the way for the concrete possibility of a brain-on-a-chip [62]. Thus, MEA technology should provide innovative solutions to keep up with the advancements in the biological counterpart. Particularly, there is a need to considerably increase channel count while being able to cope with complex signal processing strategies and to keep real-time capabilities.

MEAs with hundreds/thousands of microelectrodes and (preferably) 3D shapes are fundamental requirements for interacting with such complex biological models. Nowadays, there are different available and validated solutions for increasing the spatial resolution of MEAs for both in vitro and in vivo applications. In 2009, Berdondini and co-workers [63] developed a high-density MEA based on complementary metal-oxide-semiconductor (CMOS) technology, allowing the simultaneous acquisition from 4096 microelectrodes (21 × 21 μm^2^ sizes and a pitch of 21 μm). At the same time, Hierlemann’s group developed a MEA with 11011 planar high-density microelectrodes (7 μm diameter and placed at a pitch of 18 μm), 126 of which were able to deliver patterns of stimulation [64]. If on one hand, advancements have been made for high density MEAs, on the other, the development of 3D ones is still in a preliminary stage [65].

It is worth noticing how the same strategies used to recreate interconnected sub-populations with standard MEAs can be easily adapted to high-density devices with the natural advantage to have the possibility to design more than two interconnected regions without losing spatial information (cf. Section 3.4.1).

Regarding the high-density in vivo MEAs, recent advancements have focused on active CMOS probes [66,67,68], providing dense and extensive recording sites to isolate individual neurons across large regions of the brain. Among others, neuropixel silicon probes [66,69] are becoming widely used and enable stable recordings from thousands of sites during free behavior, even in small animals such as mice, and with a first demonstration on humans [70].

A more scalable technique involves massively parallel microwire arrays integrated with CMOS chips for neural recording [71,72,73]. These devices can sample across a plane approximately parallel to the brain surface and use arrays of parallel microwire electrodes for readout and stimulation. Among others, the Argo system offers the largest channel count microwire-based recordings in both rats and sheep to date [73]. It supports simultaneous recording from 65536 channels, sampled at 32 kHz.

#### 3.2.2. Optogenetics Recording and Stimulation

The real-time communication between neuroprosthetic devices and neurons has been classically thought and achieved by electrodes capable of detecting or inducing bio-electrical signals generated by functioning neurons and neural circuits. However, synergistic advances in optical neural bio-sensors and genetics through the last two decades within the growing field of optogenetics [74,75,76] opened a concrete possibility to establish viable real-time communications with neural systems based on optical signals [77,78].

Optogenetics was initially coined when light was used to stimulate the activity in neurons genetically modified to express light-sensitive ion channels inducing depolarizing inward calcium currents in the cells (channelrhodopsin in [79,80]; modified P2X2 receptors in this case [81]). Since then, mostly based on channelrhodopsin, halorhodopsin, and archaerhodopsin, many different optical actuators have been developed to induce or suppress the neural firing with different wavelengths and dynamical ranges in targeted neuronal populations, and the possibility of inducing a selective expression of the optical sensors in defined neural types (such as glutamatergic, GABAergic, dopaminergic neurons, and more [82]) is a major advantage of optogenetics. Currently, actuators allow for fast control of the neural activity as required for real-time communications in neuroprosthetics down to single neuron resolution [83].

In a broader sense of all-optical approaches for recording and stimulation of neurons, optogenetics has also later been related to the simultaneous use of genetically encoded optical sensors for transduction of the neural firing in fluorescence signals, through intracellular calcium variations induced by action potentials (Genetically Encoded Calcium Indicators—GECIs) or transmembrane voltage variations (Genetically Encoded Voltage Indicators—GEVIs) [84,85,86]. GECIs have been first introduced and widely used in vitro and in vivo with single-cell resolution and high signal-to-noise ratio [86]. GECIs’ main limit for real-time communication and neuroprosthetics is mainly represented by their intrinsic slow dynamics related to the intracellular calcium variations with consequent difficulty in reaching single action potential resolution [87,88,89]. GEVIs have been later introduced and represent the cutting edge of optical neural firing recording for neuroprosthetics, guaranteeing fast resolution (single action potentials) of neural firing, although with a lower signal-to-noise ratio compared to GECIs [90]. The challenges in relation to the application of GEVIs in real-time communication are mostly related to the interface and fast pre-processing of signals acquired, possibly with high-speed CMOS cameras [90].

When talking about optogenetics and fast real-time spatial-temporal resolution in neural systems, especially in in vivo conditions, an issue is the use of technologies based on two photons signals, which guarantees deep and high spatial resolution [91] but also expensive and highly challenging technological implementations in relation to freely moving conditions [92]. Notably, in recent years the use of GRIN lenses has opened new possibilities for one-photon single-cell resolution recordings and in vivo stimulations in freely moving conditions [93].

Finally, the real-time patterning of light sources is a key element to deliver information with high content to the neurons within a circuit [76]. In this context, [78] provided a clear optical real-time unidirectional communication from in silico spiking neural network (SNN) to an in vitro biological neural network (BNN) using a video-projector to generate patterned binary 8 × 8 squared micro-images for optogenetics stimulations (Figure 3). The authors quantified information transmission (IT) between the SNN and the BNN by looking at the correlation of the similarity between INPUT (i.e., the stimuli) pairs versus the similarity between OUTPUT pairs (where each OUTPUT is a vector representing the neuronal response of the BNN). The rationale is that, in a condition when information transmission occurs, two similar INPUTs to the BNN should elicit two similar OUTPUT patterns in the BNN. The authors first identified, for each experimental session of SNN-to-BNN communication, the optimal post-stimulus time window and threshold of BNN response maximizing the IT (maxIT; Figure 4a). Next, they compared the maxIT across different experimental sessions characterized by different frequency (Figure 4c) and intensity ranges of the INPUT (both controlled by the intrinsic activity of the SNN). Finally, they found that under entrainment of the BNN (Figure 4d), the linear response range (Figure 4b) of the BNN reproduced the optimal conditions for information transmission (Figure 4c,d).

Specifically, in optimal information transmission conditions, the spontaneous synchronizations in the SNN (mimicking the ones present in the BNN) entrained the global synchronizations in the BNN through optical stimulations (Figure 5). Information transmission could take place in linear responses (red dots in Figure 4b), which occurred in between the entrained (i.e., evoked) global synchronizations in the BNN (supra-linear blue responses in Figure 4b).

### 3.3. Neuromorphic Processing of Neural Signals

A key feature of bio-hybrid neuroprostheses is the real-time processing and decoding of neural signals to drive feedback for neural function modulation or replacement. A set of technological challenges are associated with this process, including the need for fast (real-time) complex computation of massive amounts of data originating from hundreds or thousands of channels and extraction of meaningful features to control stimulation. To this end, enhancement of biohybrid systems with ANN is the emerging strategy, given its capability of an adaptive and more natural interaction between the biological and artificial counterparts. These ANNs can be used for detection and classification but also for mimicking the biological neural network for replacement experiments, as discussed in the following sections.

#### 3.3.1. Detection and Classification

Real-time detection of spike and burst events, a crucial step for performing closed-loop stimulation, still represents a major challenge in the neural signal processing field. Neuromorphic computing has been used to address these issues, and at least two novel spike and burst algorithms have provided interesting results [94,95].

High-quality spike detection in extracellular field potential and generation of low-latency stimulation pulses (<10 ms) are the basic requirements to perform activity-dependent stimulation (ADS), which is a promising branch of closed-loop stimulation to enhance plasticity after brain injury [39]. In a recent work [95], we aimed at improving the original code of an open-source commercial system (Intan RHS Stim/Recording Controller), implementing a spike detection state machine on a field-programmable gate array (FPGA) to perform ADS with a latency < 1 ms. The algorithm was based on a programmable set of inclusive/exclusive thresholds L_i to reject biological artifacts potentially introduced by animal chewing and movements with the following characteristics (see Figure 6a): (i) ai stands for the voltage threshold (µV) that the waveform must cross to be identified as a spike, (ii) bi refers to the inclusive onset sample of the threshold L_i, (iii) ci refers to the exclusive end sample of the threshold L_i, (iv) di refers to the threshold polarity, if it is equal to 0 (inclusive) the signal absolute value must cross the threshold, conversely, if it is equal to 1 (exclusive) it must not cross the threshold, and (v) ei determines if a threshold L_i is involved in the decision process or not. A comparison of the in vivo spike detection state machine performance with offline detection of spikes in signal epochs containing high-amplitude artifacts highlighted the capability to trigger stimulation with a maximum delay of 967 µs and an overall accuracy between 72% and 97% [95]. Such an approach stands as a reference for closed-loop system design for ADS, overcoming historical limitations of monopolar voltage threshold algorithms [96] and providing a convenient implementation as a modification to the existing open-source code, thus avoiding the implementation of the full data acquisition circuit such as in [97,98,99,100]. Moreover, the state machine allows for remarkable flexibility in terms of threshold levels parametrization during ongoing acquisition stages, leading to higher selectivity in the spike detection.

Synchronized, packed, and distributed neural network spiking activity, known as burst, is a typical feature characterizing the neuronal signal. Given the wide range of bursting patterns, several algorithms have been developed for the detection of such events. Nevertheless, none of the currently available approaches are universally recognized as suitable to identify all the different bursting event classes. In [94], we proposed a different low-power consumption implementation method based on an FPGA neuromorphic auditory sensor (NAS). Neuromorphic signal processing and wide band signals have been adopted as a first processing step rather than standard spike detection. The in vitro electrophysiological recording was first converted into an audio signal using interpolation to reach a sampling frequency of 48 kHz. Then, spike trains were generated using the NAS spike generator. Afterwards, a frequency decomposition was performed, streaming this information using the address-event representation (AER) on a monitor. The output of the sensor was sent to a trained SpiNNaker board which discerned between bursting and non-bursting events (see Figure 6b). The comparison between this data-driven approach with conventional spike- and raw-based burst identification algorithms showed similar results in terms of number of detected events, mean duration, and correlation of the bursts. Our work represents a further example in the framework of signal processing with a neuromorphic computational approach [47,101], which extends the application spectrum of such systems to neural signal processing for spike sorting. The results obtained are consistent with those found in state-of-the-art, demonstrating how this approach can be exploited for neuroengineering purposes, thus paving the way to an innovative real-time burst detection perspective.

The emergent strategies for brain repair/replacement point to the design of neuromorphic systems capable of establishing adaptive bidirectional communication by means of a closed-loop architecture between biological and artificial components. The key factors of such systems consist of real-time decoding of neural activity, learning adaptation, miniaturization, and increased power consumption efficiency. Moreover, spiking neural networks (SNNs) represent a promising alternative in the context of reproducing brain pattern recognition feature performing online spike/burst detection and spike sorting. Indeed, the incorporation of SNNs with spike-timing-dependent plasticity (STDP) rules implemented on neuromorphic low power consumption hardware allows performing real-time unsupervised learning [102]. This may be of particular relevance in the detection and classification of electrographical events in the network level of brain organization represented in LFP signals, the viability of which remains to be investigated.

#### 3.3.2. Mimicking Biological Dynamics

The capability to compute, generate, and deliver spatiotemporal complex stimuli that have a natural dialogue with the dynamical complexity of the brain has been already demonstrated to be of paramount importance in the development of neuroprostheses. Pulsatile electrical stimulation in which the intervals between pulses follow a power law distribution with unitary exponent, thus mimicking natural like scale-free processes occurring in the brain, is capable of recruiting isolated cells [103] and small neuronal networks in vitro [104] with much higher fidelity than conventional fixed frequency. The same principle and pattern of stimulus have been used as a form of entraining epileptogenic networks into healthy oscillatory patterns, and thus to suppress epileptiform activity in animal models with negligible impact on neural function [105,106]. Other forms of non-standard patterns of stimulation correlated, in different degrees, to ongoing neurodynamics have also been successfully used in silico and in vivo investigation of therapeutic neuromodulation alternatives for brain rewiring [107,108,109], for treating Parkinsonian animals [110] and humans [111], to increase cortico-spinal excitability in an individualized fashion [112], and also in other motor disorders [113,114].

Artificial neural networks (ANN) represent the most important strategy for mimicking the complexity of neurobiological dynamics in neuromorphic-based neuroprostheses. They can be bio-inspired (i.e., computational systems inspired by biological architectures) or bio-mimetic (i.e., realistic models of living systems for real-time emulation). In the ANN family, spiking neural networks (SNN) are the most biophysically plausible once they can reproduce neuronal phenomena, starting from electrophysiological activity in single cells (soma, axon, and dendrites) up to the plasticity of the network. Three different hierarchical levels can be obtained when using SNN: (1) network level with spike timing (Figure 7a); (2) spike shape at the single cell level (Figure 7b); and (3) spatial configuration of neuron including dendritic spike, axon propagation, soma, etc. (Figure 7c). Izhikevich neurons (such as those used in Buccelli et al., 2019) are notable for being able to reproduce a wide variety of biologically realistic neuronal firing patterns by merely changing parameters of a low-complexity two-dimensional model [115]. By combining bio-inspired models with a multi-compartment approach, it becomes feasible to also model the morphological characteristics of a neuron and, consequently, design an SNN that is able to reproduce the spatial configuration of neurons and associated spike timing and morphology.

SNN-based neuroprostheses have been used for bridge experiments to restore locomotion-like activity in isolated spinal cords of rats submitted to complete transection [43], in the previously described replacement experiments [47], and has also been implemented in hardware [116]. This promising progress has also unveiled the main difficulties in the design of SNN-based neuroprostheses, which lie in the choice of the model. The complexity of the model imposed by the application has to be compared to the implementation cost induced. The choice will therefore depend on the tradeoff between complexity of the model and available hardware resources.

One mandatory requirement for these systems is the real-time bidirectional communication between BNN and the SNN. Different hardware systems can be used for this purpose, like CPU, GPU, or dedicated mixed signal electronics hardware like FPGA or ASICs. However, the real-time constraint implies that FPGA and ASICs are a more promising choice due to their parallel architecture, low latency, and low power consumption. Another way to explore SNN applications comes from the emergence of new technologies that can reproduce synapse dynamics. A promising system could be designed from the use of a hardware SNN that can both directly interface with living tissue and adapt based on biofeedback. In that case, organic synapses based on biochemical signaling activity can be a solution [117,118]. Interfacing SNN with memristors that feature plasticity [119] is also an interesting lead to explore for these neuroprostheses.

### 3.4. Towards Translation of Neuromorphic Prostheses

Thanks to the real-time processing and to the possibility to implement biomimetic strategies, according to the ‘translational methodology introduced in Section 2.3, pioneering in vitro neuroprosthetic devices, such as the one in the BrainBow project, demonstrated the ability to restore the functioning of an injured neuronal network. Nevertheless, the road towards translation is still long and requires the combined and synergistic work of experimentalists for testing innovative technological solutions in both in vitro and in vivo settings. New directions along this road are presented in the following paragraphs.

#### 3.4.1. In Vitro Experiments

When designing new generation neuroprostheses, it is fundamental to have an in vitro model that can mimic as much as possible the real structure and the functionality of the brain. Is a bi-modular cortical network a realistic model? Can bi-dimensional models recreate the in vivo microenvironment? Are primary neuron cultures the best approach? To try answering these paramount questions, different kinds of sensors (e.g., MEAs Figure 8a,b, Section 3.2.1) have been employed in the evaluation of in vitro neuronal cultures. Moreover, we have to consider that the human brain is composed of different types of neurons that differ in shape and function (heterogeneity). Furthermore, they are organized according to very precise schemes (modularity, [120], Figure 8d) and in a three-dimensional (3D) arrangement (Figure 8c).

These issues started being addressed in recent decades. By exploiting PDMS masks, some attempts have been made to recreate neuronal circuits of physiological relevance that connect neurons extracted from different brain regions (Figure 8e). A circuit of particular interest is the cortico-striatal one involved in the genesis of Huntington’s disease. Virlogeux and collaborators [121] recreated such a circuit using a two-compartment PDMS mask to quantify the contribution of pre- and post-synaptic neurons in the initial phase of the disease. The interaction between thalamus and cortex has also been the center of great interest. In vitro studies highlighted how cortical neurons act as the site of initiation of bursting events [122] and that the thalamic ones promote a redistribution of inhibitory functional connections [123]. Recently, the cortico-hippocampal circuit has been studied from both a functional and a structural point of view, pointing out the role of inhibitory connections during the development of the network in modulating the cortical dynamic [124]. In addition, these experimental models based on interacting neuronal populations allow us to perform experiments of electrical or chemical stimulation to quantify the role of the connections between the sub-assemblies and how different neuronal types respond to an external modulation [125]. The bimodular models were overcome in 2017 with an in vitro model where neurons extracted from the cortex, hippocampus, and amygdala were interconnected with the aim to understand the onset and the development of schizophrenia [126].

**Figure 8 brainsci-12-01578-f008:**
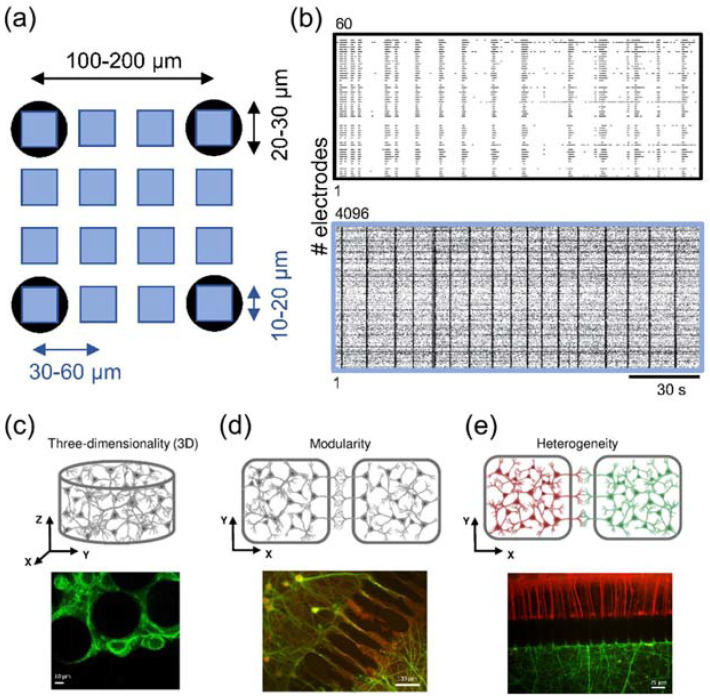
**In vitro approaches and topological features for realistic experimental models.** (**a**) High-density MEAs (blue squares) provide a higher spatial resolution of the neuronal activity than conventional solutions (black circles). The sketch shows a typical scenario where the same surface can be mapped with 16 high-density microelectrodes with respect to only 4. This feature result is fundamental if many interacting neuronal population are coupled to a MEA. (**b**) Example of spontaneous cortical activity recorded by means of a 60- (top) and 4096-electrodes (bottom) MEA. In addition to the technological part, in vitro experimental models should simultaneously embed three features: (**c**) three-dimensionality connectivity (top, adapted from [123], bottom, from [124]); (**d**) modularity topology (top, adapted from [123]); (**e**) heterogeneity, i.e., the existence of different neuronal types (top, adapted from [123], bottom, from [122], where cortical (red) and thalamic (green) neurons are interconnected to realize a complex neuronal networks).

Another critical parameter for enhancing the realism of in vitro models is the 3D microenvironment, essential for the correct expression of neural cell phenotypes [127]. In 2008, Pautot and colleagues developed a method for culturing neurons on a 3D scaffold made of layered silica beads [128]. However, such a pioneering approach lacks most of the mechanical properties of brain tissue, such as porosity and stiffness. In order to mimic the in vivo biological structure, scaffolds based on, or modified with, polysaccharides were used. Some examples are hyaluronic acid-based hydrogels [129] and chitosan hydrogels [130]. Another material widely used to design the scaffold is the collagen protein, sometimes mixed to silk [131]. Although these approaches allow reproducing in vitro some mechanical properties of the in vivo tissue, they are subject to degradation, limiting their use in predicting the development mechanisms of the neuronal network, and consequently in the evaluation of the onset and progression of diseases. A possible solution was developed in 2015 [132]: a PDMS scaffold that allowed maintaining a reasonable control on the stiffness and pore size, limiting, however, the possibility of recording the electrophysiological activity.

An application where the features of heterogeneity, modularity, and tridimensionality simultaneously coexist was presented in 2020, where the authors recreated the cortico-hippocampal circuit in a 3D fashion [62].

Finally, in the definition of a valid biological counterpart for the realization and testing of new generation neuroprosthesis, we should replace the murine-based biological tissue with human-derived cells. New in vitro systems will necessarily have to abandon animal models that show inherent genetic differences in the neuronal population [133]. A relevant alternative is offered by human embryonic stem cells (hESCs) and induced pluripotent stem cells (hiPSCs), which help in the definition of personalized interconnected 3D neuronal assemblies [134].

#### 3.4.2. In Vivo Experiments

As underlined in the previous sections, to create neuroprosthetics therapeutic solution for health care, the real-time communication between a computational device and the nervous system is mandatory. To guarantee translation of the results towards human applications, in vivo experiments, to date, still represents a necessary step. Recent work in vivo has investigated the feasibility and efficacy of intracortical microstimulation to couple the neural activity in the premotor to that of the sensory cortex to ‘bypass’ a lesion in the motor area according to a closed-loop activity-dependent stimulation (i.e., ADS) paradigm [39,135,136]. The hypothesis behind the use of a closed-loop ADS approach is rooted on the seminal work of Fetz and co-workers, who first demonstrated the possibility to promote Hebbian plasticity by artificially pairing the spiking activity of two cortical neurons in a monkey brain [38]. To enable current ADS technology with neuromorphic capabilities is the obvious next step. One can quickly realize the benefits of functionally coupling the pre-motor area (reading) to the sensorimotor cortex (writing) by means of natural-like rules of association—which is fully achievable by neuromorphic devices—in order to obtain maximal plasticity performance.

In parallel to the in vivo work, recent advancements in human neurotechnologies have allowed the design of novel adaptive deep brain stimulation (aDBS) therapies for Parkinson’s disease (PD). The amplitude (or power) of subthalamic (STN) beta band (11–30 Hz range) LFP oscillations, which is sensitive to the clinical and dopaminergic state of the patient [137,138,139], was used in this case as a biomarker for controlling DBS [140,141,142].

Indeed, the technological and scientific progress in neuroengineering and neurotechnology has allowed for the development of advanced platforms that support diverse methods for in vivo neuroscience [143]. As underlined in the previous sections, the number of recorded neurons and the number of stimulating channels, as well as the development of powerful algorithms for real-time data processing, will be crucial in the near future to develop effective neuroprostheses. To this end, the combination of the above cited recorded/stimulation techniques, together with the most recent advancements in real-time processing of neural signals, which also exploit the use of ANN/SNN [94,95], will allow to soon translate closed-loop innovative neuroprosthetic devices, first to the preclinical and then to the clinical level.

In the near future, the simulation of SNNs with an increasing number of units [144,145,146,147,148] will become necessary to cope with the complexity of the central nervous system. By real-time encoding the dynamics of an SNN, it will be possible to deliver more ‘neural-like’ stimulation patterns, by means of intracortical microstimulation of the brain tissue. This will be the first step for the realization of a complete bi-directional neuromorphic real-time system, able to exploit the SNN as miniaturized neuroprosthetic chip to perform replacement of damaged micro-circuits (i.e., thus restoring local information processing or segregation), and then capable of dialoguing within larger brain networks.

As a further step forward, the last frontier of neuroprosthetics for the brain concerns the integration of biological tissue exploiting hybridized brain tissue grafts, where the neuromorphic component of the graft assists and stimulates the integration of the biological part in the brain [21].

## 4. Conclusions

A new concept of neuroprosthesis based on neuromorphic elements, such as SNNs, has been introduced, reviewed, and described as a technological solution to restore neural functions lost because of a brain lesion, such as stroke. In this framework, the biological plausibility of the SNN does not have a mere aesthetic function but a potential functional importance. The advancements in SNNs implementation will allow, in the future, to obtain less intuitive and more complex behaviors not obtainable through classic control systems. The path, as happens with deep learning systems in image recognition, could foresee an ever-increasing level of complexity that is hardly replicable through simpler algorithms. Indeed, neuromorphic computing systems allow real-time communication and neural data processing involving a higher number of channels/electrodes/neurons. With this in mind, the high correspondence between SNNs and BNNs can be an encouraging starting point for an increasingly integrated development of hybrid systems. Furthermore, SNN can emulate the biological network at the single cell level, keeping the dynamics and the plasticity of the whole network. It is then a good candidate for replacement experiments. The road towards translation is still long, but the here-presented promising technologies and results constitute an important milestone to start from, and to open up new avenues for restoring lost capabilities in patients with disabilities.

## Figures and Tables

**Figure 1 brainsci-12-01578-f001:**
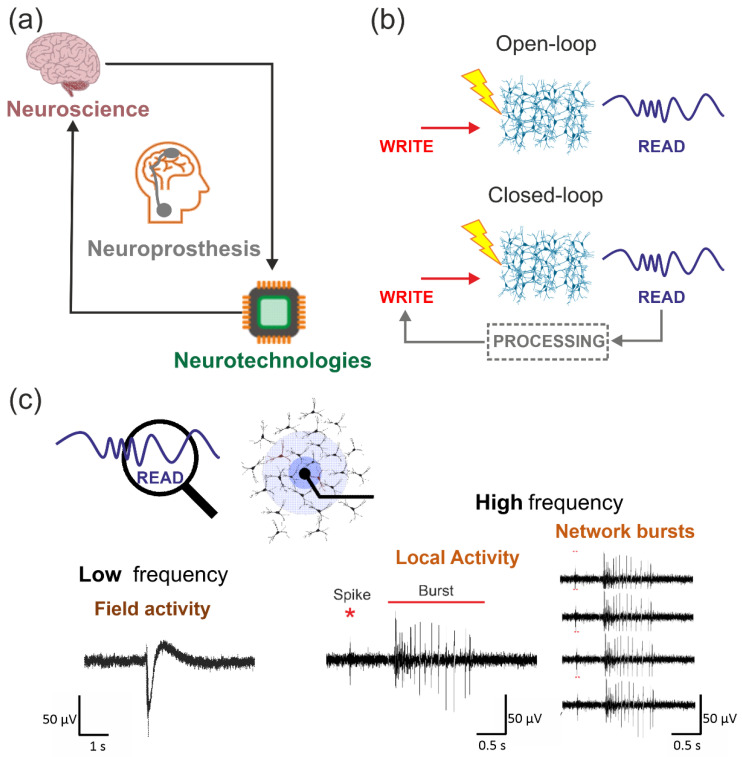
**The main elements of a neuroprosthesis.** (**a**). Neuroprostheses are the results of the dual interaction between neuroscience and neurotechnologies. (**b**). Open- and closed-loop architectures. In the open-loop modality (top), the delivered stimuli are not correlated to the network activity. On the contrary, closed-loop configurations (bottom) are based on feedback: the signals coming from the network are processed and specific features are extracted, thus producing triggering events which are responsible for delivering stimulation pulses according to the read network state. (**c**). Various options on the reading side are available for closed-loop. In the low frequency bands (up to 300 Hz), local field potentials can be detected, which carry information from a larger volume of the brain tissue and thus are more informative of the network activity. The high frequency components of the signal (>300 Hz) are characterized by fast extracellular spikes (indicated with a red asterisk—*), which represent the local activity in the neighborhood of the electrodes. Groups of spikes represent a burst (indicated by a horizontal red bar), while the presence of several bursts on the different recording electrodes generates the ‘network burst’.

**Figure 2 brainsci-12-01578-f002:**
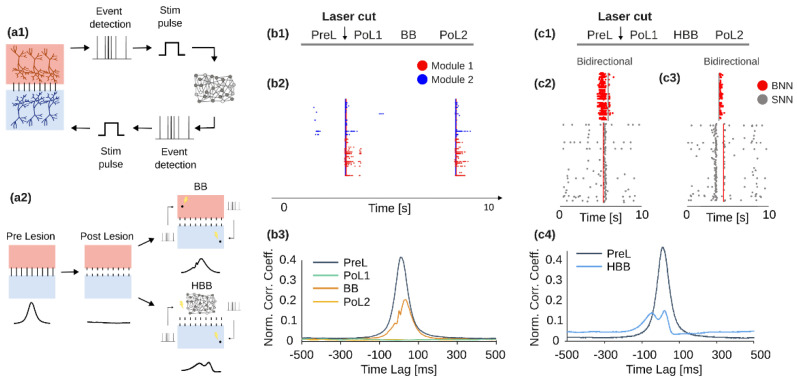
**Demonstrator of an in vitro neuromorphic prosthesis.** (**a1**) Scheme of the bi-directional interface between the biological neural network—BNN (i.e., a bi-modular culture of biological neurons over a micro-electrode array—MEA) and the spiking neural network—SNN. The actual system is composed by the commercial amplifier, hosting the MEA with the culture, and the hardware board to perform the closed-loop real-time processing (i.e., spike detection and event detection) of the biological signals. (**a2**) The cartoon of a biological bi-modular culture over a MEA is reported (*left*). A focal lesion is performed to cut the connections between the two modules (*center*). Two are the possible reconnection strategies: the *Bidirectional Bridging—BB* (*top right*, used when the connection between the two modules is lost) and the *Hybrid Bidirectional Bridging—HBB* (*bottom right*, used when, after the lesion, one of the two modules is not functioning anymore, and thus it exploits an artificial spiking neural network—SNN). (**b1**) Schematic of the BB experimental protocol: after the first phase of basal activity (PreL), a laser cut (i.e., the lesion) was performed. Then, spontaneous activity was recorded again (PoL1), followed by the BB strategy and a final phase of spontaneous activity (PoL2). (**b2**) A 10-s raster plot of the network bursting activity of one representative experiment during a BB phase. (**b3**) The cross-correlation (CC) function for one representative experiment during the four phases of the experiment (different colors). (**c1**) Schematic of the HBB experimental protocol: after the first phase of basal activity (Pre-Lesion—PreL), a laser cut (i.e., the lesion) was performed. Then, spontaneous activity was recorded again (Post Lesion1—PoL1) followed by the HBB strategy and a final phase of spontaneous activity (Post Lesion2—PoL2). (**c2**) Raster plot of a spontaneous network burst in the BNN which triggers a stimulation to the SNN, which generates a network burst and a stimulation back to the BNN. (**c3**) Opposite situation with respect panel (**c2**): a spontaneous network burst is first generated in the SNN (*right*). (**c4**) CC function for one representative experiment during preL and HBB phases (different colours). Modified from [47].

**Figure 3 brainsci-12-01578-f003:**
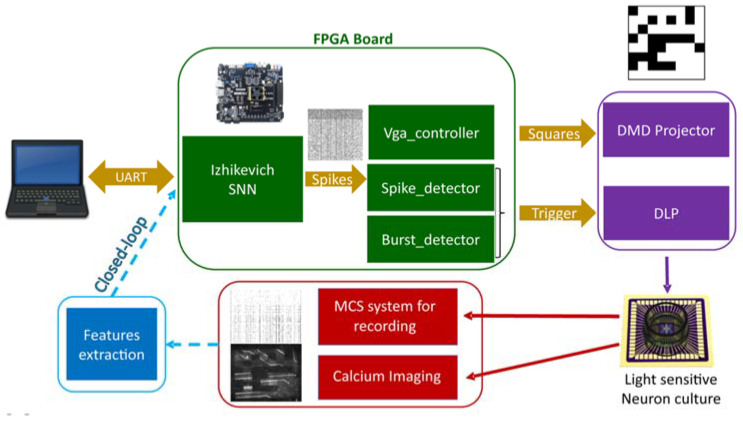
**Scheme of the experimental set-up.** An Izhikevich SNN is implemented into FPGA board. Spikes outputs are converted into 8 × 8 images through vga to a Digital Micromirror Device projector (DMD) organized around an upright microscope where a biological neuronal network cultured on a multi-electrode array is placed. Neuronal activities are recorded by both electrical signals (MCS system) and optical signals (Calcium imaging). The next step is to close the loop, adding feature extraction from biological activity to feed in the SNN to achieve a real-time bi-directional communication. Adapted from [78].

**Figure 4 brainsci-12-01578-f004:**
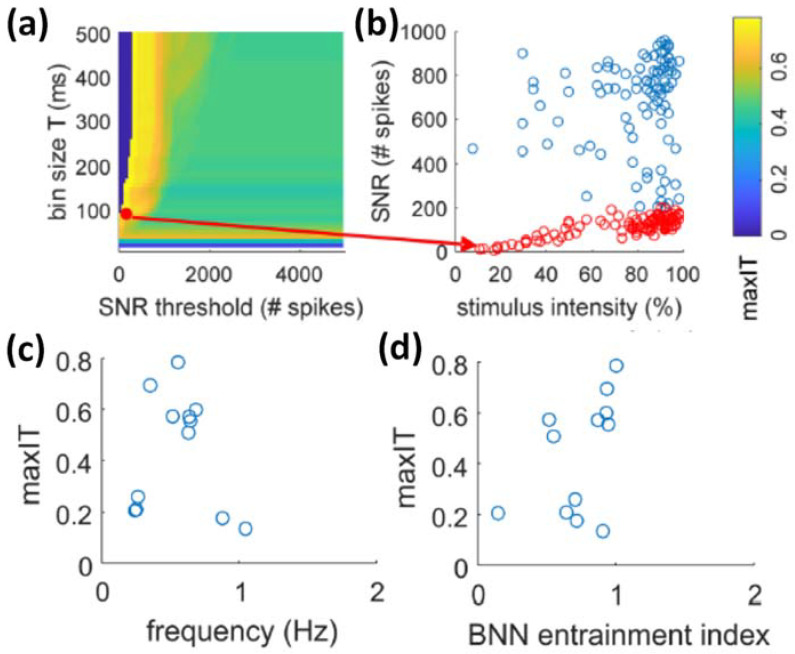
**Information transmission from SNN to BNN.** (**a**) Heatmap of the maxIT (maximum information transmission) as a function of the response bin size T and the threshold on the scalar network response (SNR). The SNR is the count of all spikes evoked in the BNN within the time T following the stimulus. The threshold on the SNR sets the maximum limit of the SNR over which responses would be discarded for the maxIT quantification. This is to discard overshooting responses linked to network synchronization dynamics and not directly due to the stimulus feature such as the intensity, as also shown in panel (**b**). (**b**) In the representative experiment, the maximal information transmission was obtained with T = 90 ms and for a threshold of 200 spikes (red dot in the heatmap of panel (**a**). In the plot of the SNR as a function of stimulus intensity, the responses below threshold are shown in red and display a linear trend. (**c**) The maximum information transmission across all experiments is shown as a function of average stimulus frequency and entrainment index of the BNN. (**d**) The BNN entrainment index was calculated as the ratio of evoked (i.e., following the stimulus) versus spontaneous (i.e., distant from the stimulation time) network synchronizations. Adapted from [78].

**Figure 5 brainsci-12-01578-f005:**
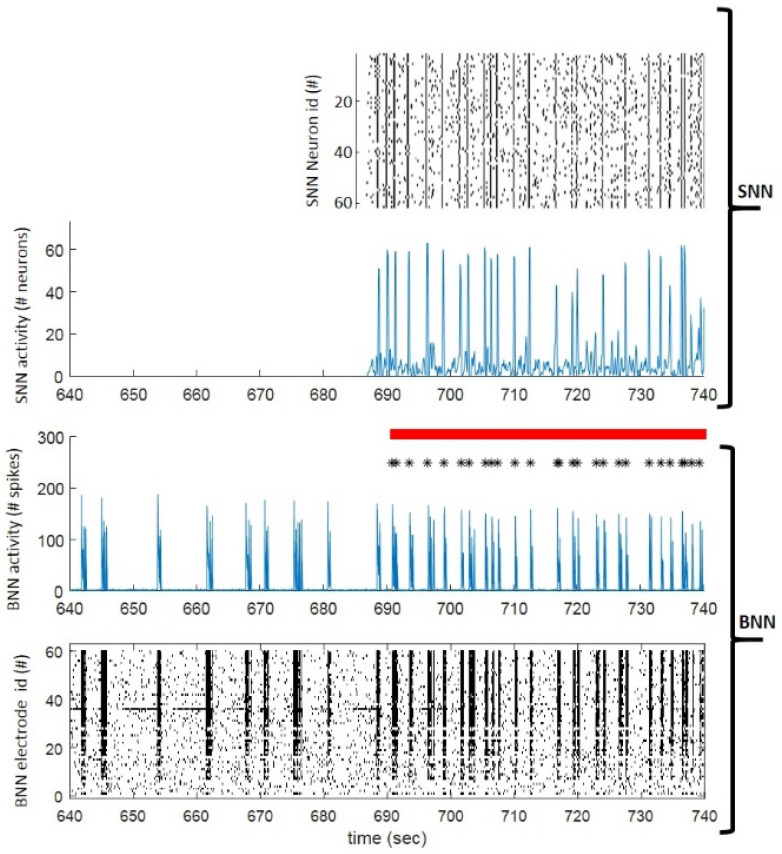
**BNN and SNN dynamics.** Focus on a few hundreds of seconds of BNN and SNN recordings. From 690 to 749 s, (red horizontal bar) the SNN dynamic trains the BNN through stimulation (black asterisks, ‘*’). The raster plots of SNN (**top**) and BNN (**bottom**) are described in time bins of a hundred milliseconds. The total number of spikes in SNN and BNN are represented by the blue plots in the same time bins as the raster plots. We can notice a high correspondence between peaks in SNN and BNN when the communication from SNN to BNN is activated.

**Figure 6 brainsci-12-01578-f006:**
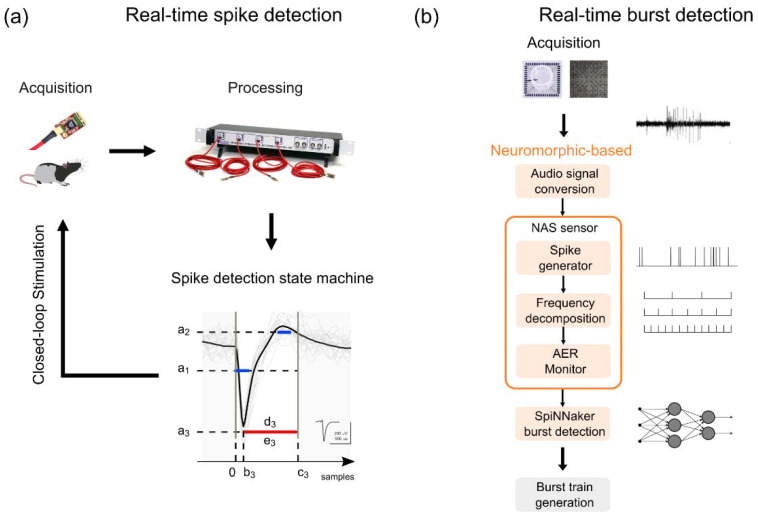
**Overview of the methodological approaches.** (**a**) Closed-loop system schematization to perform activity-dependent stimulation (ADS). The blue lines are the inclusive threshold, while the red line is exclusive. The length of the blue and red segments represents the time duration of a threshold limited by the onset sample *b_i_* and the end sample *c_i_*. The algorithm identifies as a spike a waveform characterized by an absolute value of its negative peak greater than a1 during *b*_1_–*c*_1_ and that does not exceed *a*_3_ during *b*_3_–*c*_3_, while the absolute value of its positive peak must exceed the threshold *a*_2_ during *b*_2_–*c*_2_. Modified from [95]. (**b**) Neuromorphic-based burst detection scheme. The conversion of the neuronal signal into audio allows for executing the frequency band decomposition in order to prepare the input for the SpiNNaker module to discriminate between bursting and non-bursting activity. Modified from [94].

**Figure 7 brainsci-12-01578-f007:**
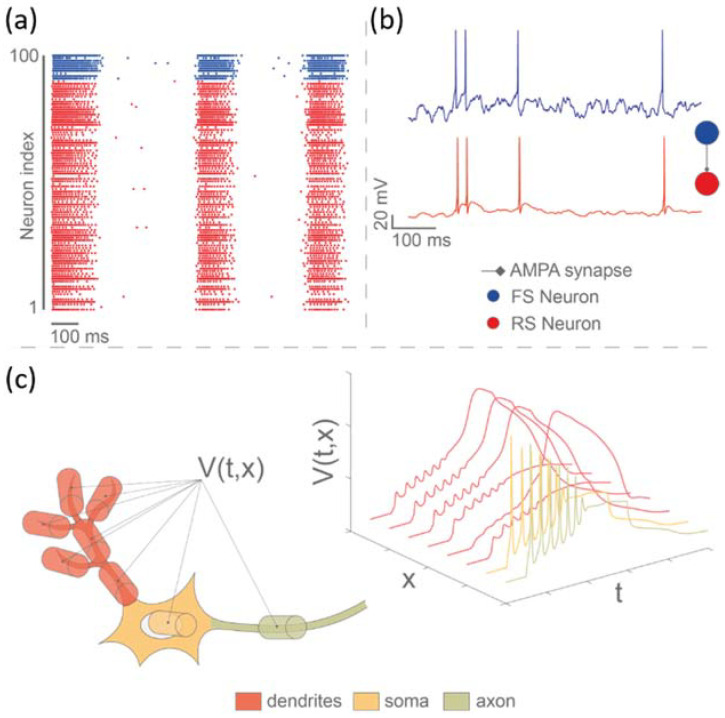
**Bio-mimetic approach for the SNN.** (**a**) Raster plot of a bio-mimetic SNN implemented on FPGA that is constituted of 100 cortical neurons distributed in a 10–90% ratio of fast spiking (FS) (blue) and regular spiking (RS) (red) neurons. FS neurons are inhibitory cortical neurons. RS neurons are the excitatory ones. The neurons are connected via synapses describing the electrical activity of AMPA and GABAa receptors. The different parameters of the model and network are tuned to obtain an activity similar to the one obtained in biology. (**b**) Membrane potential of two cortical neurons (Hodgkin-Huxley—HH model and synaptic noise) connected via AMPA synapse computed by FPGA. (**c**) Representation of the spatial structure of the neuron using compartmentalization (multicompartmental modeling).

## Data Availability

Not applicable.

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
