# Peer review of "Neuromorphic-Based Neuroprostheses for Brain Rewiring: State-of-the-Art and Perspectives in Neuroengineering"

_brainsci, 2022, doi:10.3390/brainsci12111578_

Round 1
Reviewer 1 Report
In this review article, Chiappalone et al. summarizes the latest neuromorphic approaches in the development of next generation neuroprostheses. The focus of the paper is on closed-loop technology, in which biological systems and external devices interact bidirectionally. The review introduces in detail several of the authors’ own work conducted under the European Project BrainBow and also summarizes relevant work by other groups. Overall, this is a timely and mostly well-written review on an interesting and developing topic. I have several comments for the authors to consider for possible improvement of their paper:
1. In some sections of the paper, the argument tends to focus on the authors’ own work with a lack of a comparative discussion. One example is Section 3.3.1 where the authors explain their algorithms in spike detection and burst detection. A further discussion on advantages and limitations of their approach with respect to the one by other groups would be beneficial for readers.
2. The authors provide a detailed description of their own work (reference [76]) in Section 3.2.2, but the final two paragraphs along with Figure 4 was difficult to follow. Concepts such as “maximal information transmission (maxIT)”, “optimal entrained-BNN linear conditions of maxIT”, “optimal information transmission conditions”, “threshold of the Scalar Network Response”, and “entrainment index” are impossible to follow without going into the original work. The concepts should be further explained or rephrased so that the readers can understand the points solely based on the text of this review.
3. Overall, I could not fully appreciate the essential role of SNN in neuroprostheses, beyond being “biologically plausible”. For example, why is a purely control based algorithm insufficient? Why is it important to have a high correspondence between SNNs and BNNs (Lines 431-432)?
4. The authors mention (1) scientific interests and (2) technological challenges in designing a novel (bio-hybrid) neuroprosthesis, but I missed what the authors regard as the issue in conventional neuroprostheses.
5. In the final paragraph of the Introduction (or somewhere), the authors should be explicit that the main focus of this review is on closed-loop (bidirectional) neuroprosthesis. Otherwise, the review misses many important topics in neuroprostheses, such as brain decoding for patients with motor/speech paralysis and cochlear/retinal implants.
6. Lines 136-138 are attractive introduction to closed-loop approach, which I believe is true, but should be augmented with citations(s) to literature.
7. Lines 210-211 (artificial coupling of two disconnected neuronal populations): Such experiments have also been partially realized by dynamic clamp experiments. Could be nice to mention somewhere in the text.
8. Line 312: Up to this point, in vitro neuronal networks are explained as test beds for in vivo neuroprosthesis. Please explain what the authors mean by "in vitro-based neuroprosthesis."
9. The use of the term “Optogenetics” in Lines 375-378 is misleading. Optogenetics usually refers to the genetic tools used to stimulate the transfected cells with light, but here the authors discuss GECIs and GEVIs. GECIs and GEVIs take advantage of genetically encoded proteins, but I am not aware of calling them a part of optogenetics.
10. “consequent difficulty in reaching single action potential resolution” (Line 381): This is just the opposite of what ref [85] is stating. I understand that there are arguments that not all action potentials are resolved in calcium imaging, but a more appropriate reference needs to be cited.
11. Minor points:
--Line 99: essays -> assays
--Line 133: Fig. 1a, top-> Fig. 1b, top
--Lines 330-331: A strange citation to Fig. 8a-8b. These panels should probably be separated from the rest of Fig. 8 and replaced in an appropriate position.
--Lines 333-334: metal oxides semiconductor technology (CMOS) -> complementary metal-oxide-semiconductor (CMOS) technology
--Line 335: Hirlemann -> Hierlemann
--Line 349: .. -> .
--Line 429: okit -> plot?
--Reference number should be added to the figures (figure panels) if they come from existing literature.
Author Response
In this review article, Chiappalone et al. summarizes the latest neuromorphic approaches in the development of next generation neuroprostheses. The focus of the paper is on closed-loop technology, in which biological systems and external devices interact bidirectionally. The review introduces in detail several of the authors’ own work conducted under the European Project BrainBow and also summarizes relevant work by other groups. Overall, this is a timely and mostly well-written review on an interesting and developing topic. I have several comments for the authors to consider for possible improvement of their paper:
R. We thank the reviewer for the positive comments and very insightful recommendations. We have addressed each one of them accordingly and our respective modifications of the manuscript are described separately below.
1. In some sections of the paper, the argument tends to focus on the authors’ own work with a lack of a comparative discussion. One example is Section 3.3.1 where the authors explain their algorithms in spike detection and burst detection. A further discussion on advantages and limitations of their approach with respect to the one by other groups would be beneficial for readers.
R. We agree about the lack of a comparative discussion, so we added it through the manuscript. Specifically, in Section 3.3.1, we have included multiple references to works by other groups highlighting major differences and eventual pros and cons of our approaches with respect to the literature results and methods.
2. The authors provide a detailed description of their own work (reference [76]) in Section 3.2.2, but the final two paragraphs along with Figure 4 was difficult to follow. Concepts such as “maximal information transmission (maxIT)”, “optimal entrained-BNN linear conditions of maxIT”, “optimal information transmission conditions”, “threshold of the Scalar Network Response”, and “entrainment index” are impossible to follow without going into the original work. The concepts should be further explained or rephrased so that the readers can understand the points solely based on the text of this review.
R. We thank the reviewer for pointing this out. In the revised version of the manuscript, we now provide further explanation of the mentioned concepts and replaced the “optimal entrained-BNN linear conditions of maxIT” with a devoted explanation of the associated concepts.
3. Overall, I could not fully appreciate the essential role of SNN in neuroprostheses, beyond being “biologically plausible”. For example, why is a purely control based algorithm insufficient? Why is it important to have a high correspondence between SNNs and BNNs (Lines 431-432)?
R. We thank the reviewer for the interesting question. The biological plausibility of the SNN does not have a mere aesthetic function but a potential functional importance. The advancements in SNNs implementation will allow, in the future, to obtain less intuitive and more complex behaviors not obtainable through classic control systems. The path, as happens with deep learning systems in image recognition, could foresee an ever-increasing level of complexity hardly replicable through simpler algorithms. With this in mind, having a high correspondence between SNNs and BNNs can be an encouraging starting point for an increasingly integrated development of hybrid systems. Furthermore SNN can emulate the biological network at the single cell level keeping the dynamics and the plasticity of the whole network. It is then a good candidate for replacement experiments.
The above explanation is now reported in the Conclusions of our manuscript.
4. The authors mention (1) scientific interests and (2) technological challenges in designing a novel (bio-hybrid) neuroprosthesis, but I missed what the authors regard as the issue in conventional neuroprostheses.
R. We understand the reviewer’s concern that clearly identifying a “limiting issue” of the present state-of-the-art (conventional neuroprostheses) is an efficacious approach for subsidizing the central argument of the manuscript which is related to neuromorphic neuroengineering systems. We clarify that in this manuscript we opted for an introduction of such technologies rather as a novel avenue of investigation (the ‘brain prosthesis’) with great promise, but not necessarily stemming from the limitations of others. We believe that the modified version of the manuscript convey this idea better.
5. In the final paragraph of the Introduction (or somewhere), the authors should be explicit that the main focus of this review is on closed-loop (bidirectional) neuroprosthesis. Otherwise, the review misses many important topics in neuroprostheses, such as brain decoding for patients with motor/speech paralysis and cochlear/retinal implants.
R. Agreed. The introduction has been modified accordingly.
6. Lines 136-138 are attractive introduction to closed-loop approach, which I believe is true, but should be augmented with citations(s) to literature.
R. The reviewer is right. References on the topic were added in the new version of the manuscript.
7. Lines 210-211 (artificial coupling of two disconnected neuronal populations): Such experiments have also been partially realized by dynamic clamp experiments. Could be nice to mention somewhere in the text.
R. We thank the reviewer or their useful suggestion. In the revised version o the manuscript, we added an entire new paragraph in the identified section. We put a comment in the manuscript which refers to this question, to allow for an easier identification of the changes we made.
8. Line 312: Up to this point, in vitro neuronal networks are explained as test beds for in vivo neuroprosthesis. Please explain what the authors mean by "in vitro-based neuroprosthesis."
R. Point taken. We rephrased the sentence to better convey the true meaning related to in vitro experiments.
9. The use of the term “Optogenetics” in Lines 375-378 is misleading. Optogenetics usually refers to the genetic tools used to stimulate the transfected cells with light, but here the authors discuss GECIs and GEVIs. GECIs and GEVIs take advantage of genetically encoded proteins, but I am not aware of calling them a part of optogenetics.
R. In order to avoid the confusion on this point we changed to this in the text, to make it more understandable. We put a comment in the manuscript which refers to this question, to allow for an easier identification of the changes we made.
10. “consequent difficulty in reaching single action potential resolution” (Line 381): This is just the opposite of what ref [85] is stating. I understand that there are arguments that not all action potentials are resolved in calcium imaging, but a more appropriate reference needs to be cited.
R. Although we partially agree with the reviewer based on the full content of the mentioned paper, we provided two different citations to avoid confusion. Specifically, they are “Wei, Z. et al. A comparison of neuronal population dynamics measured with calcium imaging and electrophysiology. PLoS Comput. Biol. 16, e1008198 (2020).” and “Ali, F. & Kwan, A. C. Interpreting in vivo calcium signals from neuronal cell bodies, axons, and dendrites: a review. Neurophotonics 7, 011402 (2020).”
11. Minor points:
--Line 99: essays -> assays
--Line 133: Fig. 1a, top-> Fig. 1b, top
--Lines 330-331: A strange citation to Fig. 8a-8b. These panels should probably be separated from the rest of Fig. 8 and replaced in an appropriate position.
--Lines 333-334: metal oxides semiconductor technology (CMOS) -> complementary metal-oxide-semiconductor (CMOS) technology
--Line 335: Hirlemann -> Hierlemann
--Line 349: .. -> .
--Line 429: okit -> plot?
--Reference number should be added to the figures (figure panels) if they come from existing literature.
R. We thank the reviewer for being so carrying out such careful review and pointing out these issues. We addressed all of them accordingly. Please, refer to the commented version of the new manuscript to identify changes.
As regards citation to Fig 8a, 8b, we did not modify Figure 8, but we moved the citation to the panels a and b in the Section 3.4.1 where the image is placed. Thank for pointing this out.
Reviewer 2 Report
This is a well-reviewed and organized review article. I all agree neuroprosthesis based on neuromorphic computing engineering approaches promise to be at the core of future developments for clinical applications in brain repair. This review will provide
Author Response
We thank the reviewer for the positive comments.
Reviewer 3 Report
Neuromorphic-based neuroprostheses for brain rewiring: state-of-the art and perspectives in neuroengineering (brainsci-2000246)
(Review)
Main message of the article
In the article “Neuromorphic-based neuroprostheses for brain rewiring: state-of-the art and perspectives in neuroengineering”, Chiappalone and colleagues reviewed the first case of real-time hardware neuromorphic prosthesis to restore the bidirectional communication between two neuronal populations in vitro. Perspectives on real-time interfacing and processing of neural signals together with their applications to in vitro and in vivo experimental designs and for translational purposes were also reviewed.
General Judgment Comments
The article is very well written and organized in terms of sections. Figures are clear and explicative for the reader (well done!). Title and abstract are engaging. However, authors should mention the aspect of brain rewiring in the keywords.
Major Issues
-
- Line 26, Abstract: please refer to “people with disabilities” rather than “disabled people”. Change it throughout the text.
-
- At the end of the abstract, I would suggest adding a sentence of “conclusions”.
-
- In the keywords, I would suggest adding the aspect of brain rewiring.
-
- Please make the captions of the figures self-explanatory by providing the extended version of all abbreviations used.
Author Response
The article is very well written and organized in terms of sections. Figures are clear and explicative for the reader (well done!).
R. We thank the reviewer for the very positive feedback.
Title and abstract are engaging. However, authors should mention the aspect of brain rewiring in the keywords.
R. We agree with the reviewer. The abstract was improved and the keywords were updated.
Major Issues
- - Line 26, Abstract: please refer to “people with disabilities” rather than “disabled people”. Change it throughout the text.
- - At the end of the abstract, I would suggest adding a sentence of “conclusions”.
- - In the keywords, I would suggest adding the aspect of brain rewiring.
- - Please make the captions of the figures self-explanatory by providing the extended version of all abbreviations used.
R. All points taken. Manuscript was changed accordingly and caption were carefully inspected in order to define all the acronyms.